# Metatranscriptomic insights into the dengue patient blood microbiome: Enhanced microbial diversity and metabolic activity in severe patients

**Aanchal Yadav[1,2], Priti Devi[1,2☯], Pallawi Kumari[1,3☯], Uzma Shamim[1,4], Bansidhar Tarai[5], Sandeep Budhiraja[5], Rajesh Pandey** [1,2] *

**1** Division of Immunology and Infectious Disease Biology, INtegrative GENomics of HOst-PathogEn (INGEN-HOPE) laboratory, CSIR-Institute of Genomics and Integrative Biology (CSIR-IGIB), Delhi, India, **2** Academy of Scientific and Innovative Research (AcSIR), Ghaziabad, India, **3** Indraprastha Institute of Information Technology (IIIT) Delhi, New Delhi, India, **4** Department of Biology, Ashoka University, Sonipat, Haryana, India, **5** Max Super Speciality Hospital (A Unit of Devki Devi Foundation), Max Healthcare, Delhi, India

☯ These authors contributed equally to this work.
* rajeshp@igib.in, rajesh.p@igib.res.in

**Data Availability Statement:** RNA-seq data have been deposited and are publicly available in the NCBI Sequence Read Archive (SRA) database

## Abstract

### Background

Dengue is the most re-emergent infection, with approximately 100 million new cases reported annually, yet no effective treatment or vaccine exists. Here, we aim to define the microbial community structure and their functional profiles in the dengue positive patients with varying disease severity.

### Methodology/Principal findings

Hospital admitted 112 dengue-positive patients blood samples were analyzed by dual RNA-sequencing to simultaneously identify the transcriptionally active microbes (TAMs), their expressed genes and associated pathways. Results highlight that patients with severe dengue exhibited increased microbial diversity and presence of opportunistic species (unique and core) which includes *Bacillus cereus*, *Burkholderia pseudomallei*, *Streptococcus suis*, and *Serratia marcescens*. The functional profile analysis revealed enriched metabolic pathways such as protein degradation, nucleotide biosynthesis, ion transport, cell shape integrity, and ATP formation in severe cases, indicating the high energy demands and adaptability of these microbes.

### Conclusion

Our metatranscriptomic approach provides a species-level characterization of blood microbiome composition and reveals a heightened diversity of TAMs in patients with severe dengue, underscoring the need for further research into the role of blood microbiota in disease progression. Comparing the microbial signatures across the severity classes early in the disease offers unique potential for convenient and early diagnosis of dengue infection.

under BioProject ID: PRJNA1071729. All relevant data are within the manuscript and its Supporting Information files. The source codes for microbial gene analysis performed in the paper is freely available at https://github.com/pallawikumari/-Metagenome.

**Funding:** This study received financial support from Bill and Melinda Gates Foundation (BMGF), (Grant number - INV-033578) and Rockefeller Foundation, (grant number—2021 HTH 018) awarded to RP. PK received their salary from BMGF project. US received her salary from Rockefeller Foundation project. AY and PD acknowledge Council of Scientific & Industrial Research (CSIR), India for the fellowship support. The funders had no role in study design, data collection and analysis, decision to publish, or preparation of the manuscript.

**Competing interests:** The authors have declared that no competing interests exist.

## Author summary

Dengue poses a significant public health challenge in tropical and subtropical regions. The lack of a specific treatment or broadly effective vaccine highlights the critical need for novel approaches. In this study, we investigated the transcriptionally active microbial species (TAMs) in 112 dengue-positive patients from MAX Hospital in Delhi, India, to identify their correlations with disease severity. RNA-seq analysis revealed an increased abundance of opportunistic TAMs in severe dengue patients, with some species showing significant associations with platelet counts and total leukocyte count (TLC). Our findings underscore the underappreciated importance of the blood microbiome in dengue, particularly the heightened metabolic activity of these opportunistic species in severe cases. This study offers valuable insights for early infection diagnosis for dengue management. The species-level data identified here could pave the way for systematic exploration of microbial involvement in dengue virus (DENV) infections, potentially serving as an indicator for new adjuvant treatment strategies.

## Introduction

A holistic approach to infectious diseases is necessary to understand the biology of the disease, instead of considering it as a constant confrontation between two separate worlds, the microbial and the human. Dengue fever (DF), caused by the flavivirus dengue virus (DENV)–a single-stranded positive-sense RNA virus–is a prime example of such an infectious disease. As the most prevalent arboviral disease globally, the disease's aetiology is well known. DF is distributed worldwide in the tropics and is characterized by the rapid onset of fever in combination with mild flu-like symptoms to severe headache, rash, vascular permeability, and shock [1]. In 2022, India reported 233,251 dengue cases with 303 deaths, according to the National Center for Vector Borne Disease Control (NCVBDC). Delhi alone recorded 10,183 cases and 9 deaths in 2022, following 13,089 cases and 23 deaths in 2021 [2]. However, dengue remains classified as a neglected tropical disease of significant global concern, with currently no specific vaccine or antiviral treatment available. Despite its potential to progress to severe clinical manifestations, the lack of clinically usable biomarkers poses a challenge in predicting which patients will develop severe dengue.

In this regard, the host transcriptional responses to DENV infection have been extensive, revealing a robust host inflammatory immune response. Devignot et. al revealed an increased abundance of anti-inflammatory and repair/remodeling transcripts in dengue shock syndrome (DSS) patients [3]. Also, the presence of four antigenically related serotypes of DENV and structural heterogeneity of dynamic virions has prompted numerous studies aimed at understanding the viral determinants underlying severe dengue [4]. Besides the factors mentioned earlier, the microbiome—a complex collection of microorganisms within the body—plays a crucial role in shaping host phenotypes. Humans harbor diverse microbial communities that have been shown to influence infection susceptibility and protection [5]. Recent studies of the mosquito microbiome, the primary vector of Chikungunya, Dengue, and Zika viruses, have shown a high diversity of bacteria and viruses affecting mosquito development and physiology [6,7]. However, functional metagenomic studies in humans, secondary hosts for the dengue virus, are limited, leaving the role of microbes in modulating dengue disease severity largely unexplored [8].

To address these gaps in knowledge given the high disease load in India, we employed genomics based RNA-sequencing (RNA-seq) approach on 112 blood samples from hospital admitted dengue-positive patients. We categorized our patient cohort into mild, moderate and severe based on the platelet and total leukocyte counts, the major predictive parameters of DF [9]. The functional metagenomics analysis profiled the abundance and diversity of the early transcriptionally active microbial species associated with varying dengue disease severity. We employed BLAST-based analysis to study active microbial genes and pathways, aiming to discern differences among dengue patient severity subgroups.

## Methods

### Ethics statement

The study was designed in accordance with the Declaration of Helsinki and was approved by the institutional ethics committee of Council of Scientific & Industrial Research-Institute of Genomics and Integrative Biology, Delhi, India (Ref No: CSIR-IGIB/IHEC/2020-21/01). The patients/participants provided their written informed consent before participation in this study.

### Study design

This study was conducted at the CSIR-Institute of Genomics and Integrative Biology (IGIB), Delhi, India. Sample collection was performed by the paramedical team at the MAX Healthcare Hospital, Delhi, India, on the reporting day at the hospital by the patients. Clinical blood samples were obtained from 112 individuals testing positive for dengue during the peak dengue season in Delhi, spanning from August-November 2022. All clinical data were obtained retrospectively by reviewing the electronic medical records of each participant.

Based on the complete blood count (CBC) reports of the patients, the study categorized the 112 NS1-positive dengue patients into three groups: Dengue without Warning Signs (45 patients with normal platelet and leukocyte counts), Dengue with Warning Signs comprising two subgroups– 46 patients with normal platelet but decreased leukocyte counts (Leukopenia) and 21 patients with both low platelet counts (thrombocytopenia) and decreased leukocyte counts (Leukopenia). None of the patients exhibited severe bleeding, organ failure, or abnormal liver parameters, in accordance with the World Health Organization (WHO) dengue classification and management scheme. Consequently, a separate group for Severe Dengue was not formed. For clarity, the groups are designated as mild (no thrombocytopenia and Leukopenia), moderate (no thrombocytopenia but Leukopenia), and severe (thrombocytopenia and Leukopenia) dengue cases.

### Method details

Table 1 summarizes the reagents and resources used for carrying out the downstream mentioned protocols.

### Nucleic acid isolation

RNA extraction from the blood samples was carried out using the Qiagen QIAamp RNA Blood Mini Kit, following the manufacturer's instructions with specific modifications. The incubation and centrifugation time during the erythrocyte lysis step were reduced to 5 minutes. Additionally, for enhanced RNA purification, a 2–3 minutes' incubation period was introduced during all washing steps. The purity of the isolated RNA was assessed using a

**Table 1. Reagents, software, and algorithms used in the study with their sources for result reproduction.**

| REAGENT or RESOURCE | SOURCE | IDENTIFIER |
|---|---|---|
| Blood RNA extraction | QIAmp RNA Blood mini kit, Qiagen | Cat. No. 52304 |
| TruSeq Stranded Total RNA Library Prep Globin | Illumina | Cat. No. 20020612 |
| AMPure XP | Beckman Coulter | Cat. No. A63881 |
| Agencourt RNAClean XP Kit | Beckman Coulter | Cat. No. A63987 |
| Qubit dsDNA HS Assay kit | Symbio (Thermo Fisher Scientific) | Cat. No. Q32854 |
| Agilent 2100 Bioanalyzer | Agilent | Cat. No. 5067–4626 |
| Deposited data | | |
| RNA-seq data | This study, NCBI Sequence Read Archive (SRA) database | BioProject ID: PRJNA1071729 |
| Software and algorithms | | |
| bcl2fastq | NA | GitHub—brwnj/bcl2fastq: NextSeq specific bcl2fastq2 wrapper. |
| FastQC | [10] | Babraham Bioinformatics—FastQC A Quality Control tool for High Throughput Sequence Data |
| Trimmomatic v0.39 | [11] | USADELLAB.org—Trimmomatic: A flexible read trimming tool for Illumina NGS data |
| HISAT2 | [12] | GitHub—DaehwanKimLab/hisat2: Graph-based alignment (Hierarchical Graph FM index) |
| Samtools | [13] | https://github.com/samtools/ |
| Kraken2 | [14] | https://github.com/DerrickWood/kraken2/wiki |
| Bracken2 | [15] | https://github.com/jenniferlu717/Bracken |
| Vegan & Phyloseq | [16,17] | https://joey711.github.io/phyloseq/tutorials-index.html |
| KrakenTools | [18] | https://github.com/jenniferlu717/KrakenTools |
| metaSPAdes | [19] | https://github.com/ablab/spades |
| BLASTX | [20] | https://ftp.ncbi.nlm.nih.gov/blast/executables/blast+/LATEST/ |
| ShinyGO v0.80 | [21] | http://bioinformatics.sdstate.edu/go/ |
| STAMP | [22] | https://beikolab.cs.dal.ca/software/STAMP |

NanoDrop Microvolume Spectrophotometer and confirmed by running it through an agarose gel. Subsequently, the RNA was stored at -80°C until thawed for RNA-seq library preparation.

## Dual RNA-seq library preparation and sequencing

Libraries were prepared with a total of 250 ng of total RNA using the Illumina TruSeq Stranded Total RNA Library Prep Globin (Illumina, Cat. No. 20020612). Initial steps involved the depletion of globin mRNA and ribosomal RNA (both cytoplasmic and mitochondrial), as these two forms of abundant RNA are present in high levels in whole blood. Cleaved RNA fragments underwent first strand cDNA synthesis through reverse transcriptase and random primers. Subsequently, double-stranded cDNA was synthesized using DNA polymerase 1 and RNase H, followed by purification using AMPure XP (Beckman Coulter, A63881). Adenylation at the 3′ blunt end of double-stranded cDNA was performed, and sequencing libraries were uniquely indexed and enriched through PCR amplification.

Library quality was confirmed through size analysis on an Agilent 2100 Bioanalyzer, and library concentrations were determined using the Qubit double-stranded DNA (dsDNA) high-sensitivity (HS) assay kit (Thermo Fisher Scientific; catalog no. Q32854). Libraries were then diluted to 4nM and combined in an equimolar fashion, with 24 samples per library pool. Paired-end, 2×151 read length sequencing was performed on a NextSeq 2000 platform (Illumina) at a final loading concentration of 650 pM.

## Data analysis

### Metatranscriptomic analysis

Basecalls were converted to the raw reads (fastq format) using bcl2fastq. After that, raw reads were used to do downstream sequence analysis for quality filtering and trimming using Trimmomatic v0.39 [11] to remove adaptor and low-quality sequences. Reads were aligned to human reference genome sequence GRCh38 using HISAT2 [23] and host RNA reads were separated and extracted all the reads which could not match to human (AKA), non-human reads using samtools. Non-human reads were then used to investigate microbial taxonomic analysis using kraken2 and bracken2.

### Microbial mapping and taxonomic classification

Following these steps, Kraken2 [14], a fast taxonomic memory efficient classifier that associates short reads (k-mers) with the taxa which is the lowest common ancestor (LCA). The prebuilt database was downloaded which was constructed from the complete refseq of bacteria, archaea and viral libraries. The kraken2 function was used to run the filtered reads against this database and assign taxonomy. Bracken2 (Bayesian Reestimation of Abundance with KrakEN) [15] uses the bayesian algorithm along with the classified taxonomy to re-estimate the species level abundance by assigning the most accurate species occurring in the same sample. The Kraken2 database was used to create a Bracken-compatible database using the brackenbuild function. The Kraken2 report files for each sample were computed against the Bracken database using the bracken function and classified for each perfect reads for the phylum, genus and the species level information. The output reports were then joined using an in-house python programme and normalized and downstream analysis was done.

Before downstream analysis, the essential step is to normalize sequencing depth differences among samples to reduce the skewness in the analysis and interpretation. In our data, we have used CSS (cumulative sum scaling) on read count generated from bracken report files to make data comparable and that results are robust. Count reads were converted into a biom file using the kraken-biom function and imported into R for diversity analysis and visualization. R (v4.2.0) package phyloseq and vegan were used for Alpha and Beta diversity analysis, to further explore into data permutational multivariate analysis of variance was used to check the significance across data using bray method using adonis package in R. Alpha diversity (within-sample diversity) indices, including Shannon, Simpson and Chao-1 index, were calculated on the basis of the species profile for each sample. Beta diversity (between-sample diversity) was calculated with bray-curtis distance method using abundance data using R-package phyloseq and visualized by Principal Coordinate Analysis (PCoA) plot. Data was exported as csv files and formatting, plotting and visualization was performed in ggplot2, ggpubr, python (3.6) using the seaborn library (0.12.2) and Matplotlib (3.1) built on NymPy (1.11.0). The full analysis pipeline can be accessed through our published Star Protocols [24].

### Microbial gene analysis

All the reads specific to the bacteria were extracted from the KrakenTools using extract_kraken_reads.py script for checking the gene expression of that particular bacteria. Firstly, contigs were formed using metaSPAdes by assembling extracted reads. Reference sequences were downloaded from uniprot along with gene information for the specific microbes and a database was built for blast run using makeblastdb. BLASTX was then executed with the cutoff of e-value less than 1e-50 for better match/high quality match. Gene selection based on e-value, no gap openings, and a minimum percent identity (pident) of 50% was selected. To explore

the functional profile of the species-specific genes, pathway analysis was performed using ShinyGO, a web-based tool [21]. The source codes for microbial gene analysis performed in the paper is freely available at https://github.com/pallawikumari/-Metagenome.

## Statistical and correlation analysis

For alpha diversity, kruskal wallis test was applied and for Beta diversity analysis, PERMANOVA was calculated to determine the statistical significance using R-package, adonis. Significance of common species was checked using STAMP (statistical analysis of metagenomic profile) software with ANOVA (analysis of variance) post-hoc test Tukey-kramer with 95% confidence interval and p value $\leq 0.05$. Pearson correlation analysis was performed to investigate the correlation between microbial species and clinical features using a stat function "Corr" from R-package. A p-value $\leq 0.05$ was considered significant.

## Results

### Dengue-positive cohort and blood microbiome study design

We recruited 112 hospital reported Dengue-positive patients, all of whom tested positive for the NS1 antigen, to explore and elucidate if blood microbiota changes were associated with dengue disease severity. The samples were collected between August and November 2022 from Max Hospital in Delhi, India at the time of hospital visit. Based on the clinical parameters, Dengue-positive patients were categorized into three dengue disease severity groups: mild (n = 45), moderate (n = 46), and severe (n = 21). According to the World Health Organization (WHO) guidelines, the standard classification based on Dengue disease severity includes Dengue without Warning Signs, Dengue with Warning Signs, and Severe Dengue. However, not all patients in our study conformed to these specific classifications which are important for localization of global health problems. Patients meeting the criteria for Dengue without Warning Signs, with normal platelet and leukocyte counts, were classified as *mild cases* in our cohort. Patients with Dengue with Warning Signs were further subdivided into *two groups*: those with normal platelet counts but with leukopenia, and those with both thrombocytopenia and leukopenia. Although none of the patients precisely fit the severity categories as per WHO criteria, we designated these groups as *moderate* and *severe*, respectively, for analytical purposes.

For this study, a total RNA-seq library was prepared from the blood-isolated RNA, followed by sequencing on the Illumina NextSeq 2000 platform, as illustrated in **Fig 1**. This sequencing process yielded a total of 18784780 reads. After a thorough quality check, these reads were categorized into human-mapped reads (17159378) and unmapped ones (1625402). The differential proportions of human and non-human reads found in each group were as follows: mild (91.4% human vs 8.6% non-human), moderate (91.3% human vs 8.7% non-human), and severe (91.5% human vs 8.5% non-human). The unmapped reads were then used for downstream analysis to map against microbial data which included bacteria, archaea, and viruses. As a result, we observed an almost uniform distribution of microbial reads across the mild, moderate, and severe patients, accounting for 14.7%, 17.7%, and 17.3% respectively. Surprisingly, the average dengue viral reads were higher in mild patients compared to both moderate and severe (Mild: 22004.8, Moderate: 4472.8, Severe: 5006.3).

### Clinical parameters across the disease severities

We collected an extensive set of metadata variables from each patient. Amongst the 112 patients, the average age was 28.5 years, and 68.8% patients were male. The detailed

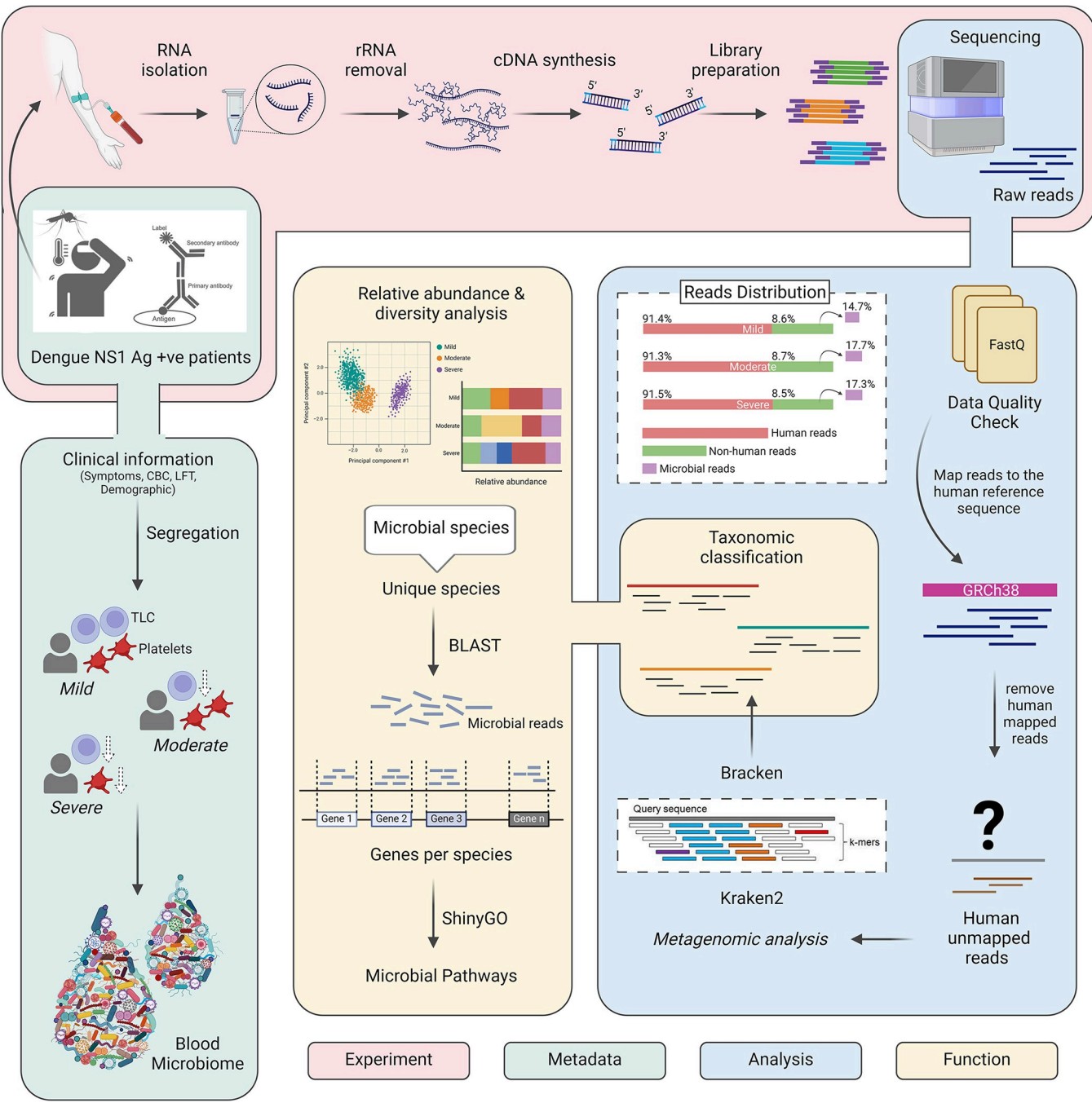

**Fig 1. Study Design and Workflow of Dengue Patient Cohort.** The schematic illustrates the experimental process for analyzing blood samples from Dengue patients classified into mild, moderate, and severe groups. RNA from these samples was subjected to dual RNA-seq using the Illumina NextSeq 2000 platform. The Figure outlines the workflow, including metatranscriptomic data analysis, which integrates functional exploration of TAMs, gene expression, and metabolic pathways. Clinical sample metadata is incorporated to provide comprehensive insights into the study findings. Figure created with Biorender.com.

demographic information alongside clinical parameters are shown in **Table 2**. Mild symptoms, including fever, chills, vomiting, diarrhoea, headache and nausea, were commonly distributed among all the three severity subgroups, with additional occurrences of orbital pain, rashes, sore throat, and constipation, observed in few cases (**Fig 2A**). Our data revealed no symptoms

**Table 2. Demographic and laboratory findings of dengue-positive patients.**

| | Dengue Clinical Parameters | Mild (n = 45) | Moderate (n = 46) | Severe (n = 21) | p-values | |
|---|---|---|---|---|---|---|
| Age | Age | 27 (18–33) | 26 (19–31.75) | 35 (18–43) | 0.14 | b |
| Gender | Gender F\|M | 15\|30 | 17\|29 | 4\|17 | 0.34 | a |
| NS1 Ag | NS1 antigen | 3.22 (0.83–3.60) | 3.5 (0.86–3.5) | 3.5 (1.9–3.5) | 0.75 | b |
| CBC | Total Leucocyte Count (TLC) | 5 (4.6–5.9) | 3 (2.5–3.575) | 2.7 (2.3–3.1) | < .001 | b |
| | RBC Count | 4.76 (4.51–4.99) | 4.835 (4.58–5.02) | 4.78 (4.51–5.58) | 0.76 | b |
| | Packed Cell, Volume | 41.2 (40.1–42.8) | 42.25 (38.4–44.22) | 43.9 (40.4–46.3) | 0.18 | b |
| | Platelet Count | 175 (155–202) | 160 (150–165) | 125 (112–135) | < .001 | b |
| | Neutrophils | 75.9 (66.7–79.5) | 63.3 (58.1–68.65) | 62.6 (57.3–73) | < .001 | b |
| | Lymphocytes | 14.1 (9.4–21.3) | 24.5 (20.3–31.975) | 26 (15.8–32.7) | < .001 | b |
| | Monocytes | 9.8 (8–12) | 10 (7.85–11.475) | 9 (6.9–11) | 0.32 | b |
| | Eosinophils | 0.2 (0–0.725) | 0.2 (0.1–0.875) | 0.5 (0.1–1) | 0.32 | b |
| | Basophils | 0.5 (0.3–0.6) | 0.6 (0.225–0.8) | 0.5 (0.3–0.8) | 0.69 | b |
| | Hb | 13.6 (12.75–14.3) | 13.9 (12.52–14.5) | 14.1 (13.1–15.3) | 0.28 | b |
| LFT | Total Protein | 7.3 (7–7.6) | 6.7 (6.5–7.3) | 6.7 (6.05–7.125) | **0.03** | b |
| | Albumin | 4.4 (4.05–4.5) | 4 (3.8–4.15) | 4 (3.675–4.1) | **0.05** | b |
| | Globulin | 3.1 (2.95–3.3) | 3 (2.65–3.15) | 2.65 (2.3–2.925) | 0.06 | b |
| | A.G. ratio | 1.4 (1.3–1.55) | 1.3 (1.2–1.5) | 1.55 (1.375–1.6) | 0.19 | b |
| | Bilirubin (Total) | 0.7 (0.55–0.8) | 0.4 (0.3–0.55) | 0.75 (0.5–0.9) | **0.02** | b |
| | Bilirubin (Direct) | 0.135 (0.12–0.17) | 0.09 (0.07–0.13) | 0.17 (0.13–0.23) | **0.02** | b |
| | Bilirubin (Indirect) | 0.56 (0.43–0.615) | 0.31 (0.24–0.42) | 0.55 (0.39–0.71) | **0.03** | b |
| | SGOT- Aspartate Transaminase (AST) | 41 (34.5–76) | 59 (44–80.5) | 97.5 (74.25–141.5) | **0.03** | b |
| | SGPT- Alanine Transaminase (ALT) | 42 (20–95) | 41 (25.5–59) | 66 (51.75–80.25) | 0.13 | b |
| | AST/ALT Ratio | 0.93 (0.82–1.725) | 1.48 (1.13–1.95) | 1.705 (1.2–2.17) | 0.08 | b |
| | Alkaline Phosphatase | 67 (60–89) | 80 (59.5–100) | 82.5 (59.75–124.25) | 0.61 | b |
| | GGTP (Gamma GT), Serum | 37 (24.5–46.5) | 19 (14–41) | 49 (32–82) | 0.08 | b |

a-chi square; b-kruskal wallis

exclusive to the severe group, with only a few overlapping symptoms between the mild-moderate and moderate-severe groups. The Venn diagram (**S1 Fig**) visually represents the overlapping and unique symptoms within the patient population. Upon thorough evaluation of clinical parameters associated with dengue, no statistically significant disparities were found in age, gender and other above-mentioned symptoms amongst the severity groups (**Table 2**). This observation indicates the absence of age and gender-related biases in the subsequent analyses. Remarkably, the NS1 antigen, suggested as a potential marker for predicting severe dengue, did not show significant variation across the groups [25].

Furthermore, in light of Dengue fever's known impact on blood and liver parameters, first we undertook a comprehensive investigation of the complete blood count (CBC) for the 112 patients, which encompass 10 parameters (**Fig 2B**). These parameters include Total Leucocyte count (TLC), Packed Cell Volume, Red blood cell count (RBC), Platelet Count, as well as Neutrophils, Lymphocytes, Monocytes, Eosinophils, Basophils, and Haemoglobin (Hb). To identify parameters associated with the severity groups, we applied statistical approaches. Within the CBC, we observed significant differences in TLC, Platelet count, Neutrophils, and Lymphocytes across the different patient groups. Specifically, TLC, Platelet count, and Neutrophils were significantly elevated in the mild cases compared to the moderate and severe patients. Consequently, thrombocytopenia and leukopenia were observed in severe Dengue patients (**Fig 2Ci-iii**). Conversely, Lymphocyte levels were significantly elevated in moderate and

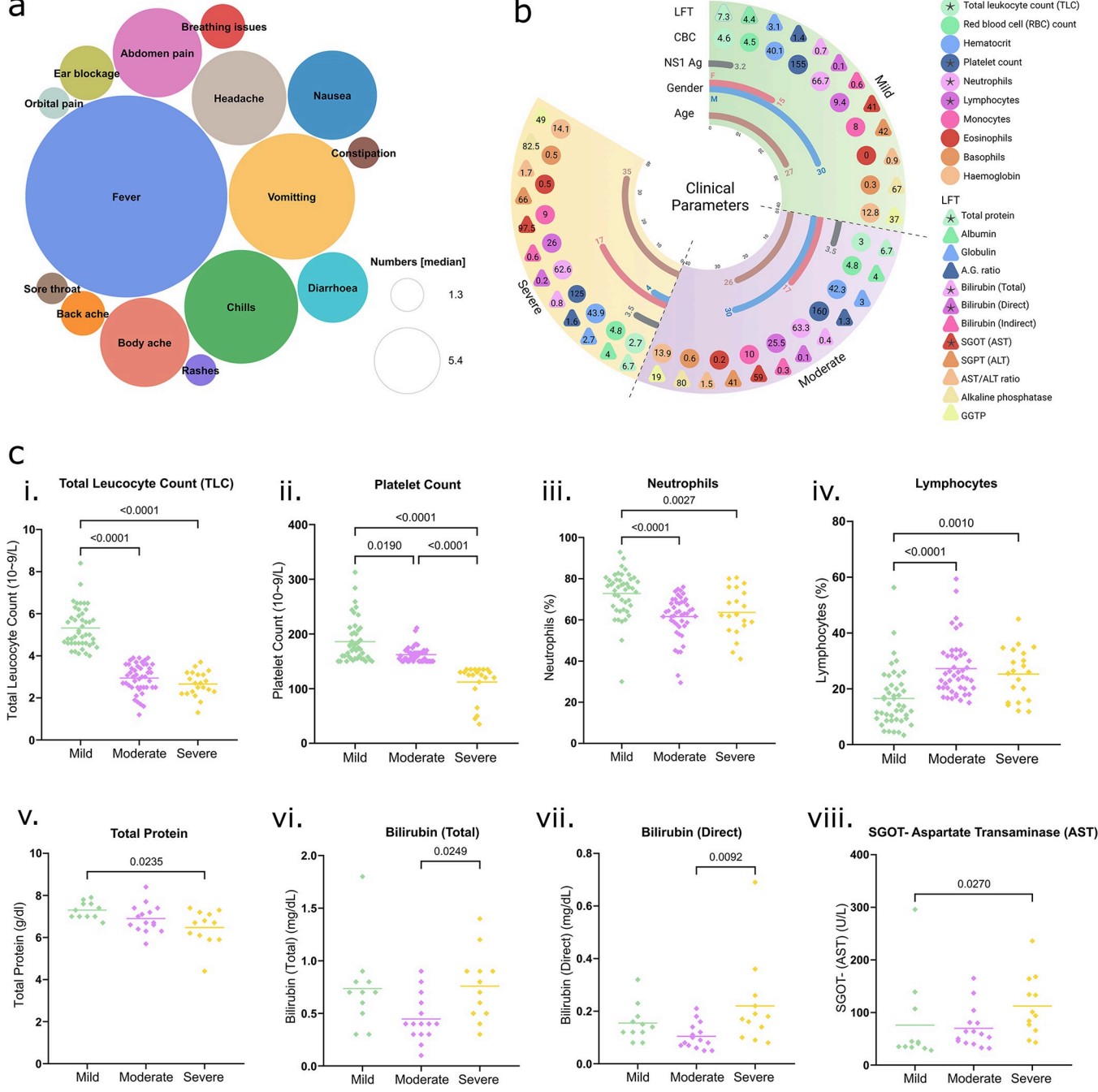

**Fig 2. Clinical Data Overview. (a)** Circle packing plot illustrates the diversity of symptoms among 112 Dengue patients. **(b)** Detailed presentation of various clinical parameters including liver function tests (LFT), complete blood counts (CBC), age, NS1 antigen status, and gender. Median values for LFT and CBC are shown, with circles representing CBC values and triangles indicating LFT values for each group. **(c)** Significant CBC parameters: **i)** Total leukocyte count, **ii)** Platelet count, **iii)** Neutrophils, **iv)** Lymphocytes, and liver parameters: **v)** Total protein, **vi)** Bilirubin (total), **vii)** Bilirubin (direct), **viii)** SGOT/AST, are shown across the three groups. Statistical significance was measured using the Mann-Whitney U test, with a p-value < 0.05 indicating significance.

severe cases compared to mild cases (**Fig 2Civ**). The normal ranges of CBC parameters are detailed in **Table 2**.

Additionally, our examination of liver parameters available for a subset of 38 patients included 12 metrics (**Fig 2B**): Total Protein, Albumin, Globulin, Albumin:Globulin ratio,

Bilirubin (total and direct), SGOT—Aspartate Transaminase (AST), SGPT—Alanine Transaminase (ALT), AST:ALT ratio, Alkaline Phosphatase, and Gamma GT (GGTP). The normal ranges for these parameters are provided in **Table 2**. Notably, we observed elevated levels of bilirubin and SGOT in severe Dengue cases (**Fig 2Cv-viii**), indicating liver damage. These clinical parameters support the severity-based classification of Dengue patients. Considering the disparity in clinical parameters despite the similar level of NS1 antigen across patients, we were prompted to investigate the microbial dynamics in patients to determine if they play a non-canonical role in defining disease severity.

## Community composition of blood microbiomes associated with Dengue disease

The influence of microbiota in dengue virus infection is still poorly understood. In our previous studies, we identified specific microbial species associated with COVID-19 severity. Here, we used the dual RNA-seq data to characterize the potential blood microbiome alterations in 112 patients with varying severities of dengue disease (mild, moderate and severe). In total, we detected ~5500 transcriptionally active microbes (TAMs) utilizing the meta-transcriptome analysis pipeline with the help of kraken2 and bracken2 tools. The identified microbial species encompasses the bacteria (3805, 68.75%), viruses (1514, 27.36%), and archaebacteria (216, 3.9%). Notably, alpha diversity, measured using the Shannon diversity index, Simpson's index, and Chao 1 estimator, exhibited no notable variances across the severity subtypes (**Figs 3Ai and S2**). To evaluate if microbiota changes correlated with disease severity, we measured the beta diversity (differences between the samples) using Bray-Curtis index. PERMANOVA analysis revealed significant differences among the severity sub-phenotypes ($p = 0.021$ and $R2 = 0.031$). PCoA of beta diversity demonstrated distinct clustering of samples between mild, moderate, and severe dengue-positive patients, indicating variability in microbial diversity, based on the infection severity (**Fig 3Aii**).

Comparing the blood microbiota profiles at phylum and genus levels across the three severity groups of Dengue patients, overall similar microbial taxa were identified. Proteobacteria (36.8%, 36.2%, 37%) followed by Firmicutes (15.8%, 16.1%, 15.3%), and Actinobacteria (15.5%, 14.6%, 14.9%) were discovered as the most abundant phyla in the blood from mild, moderate and severe Dengue patients (**Fig 3Bi**). Phyla Firmicutes was significantly abundant in the moderate with Nucleocytoviricota, which is significantly abundant in moderate and severe when compared to mild. Upon examination at the genus level, we observed that the distribution of most genera was uniform across the three groups, except for Lactobacillus and Corynebacterium, which exhibited higher abundance (>1%) in mild and moderate cases compared to the severe. Conversely, the genus *Mycobacterium* was found to have >1% relative abundance in the severe group compared to the mild and moderate groups (**Fig 3Bii**). Towards comprehensive understanding, all the details regarding phyla and genera are provided in the **S1 File.** The observed microbial community composition was then delved deeper, albeit cautiously as it is being inferred from the RNA-seq data, into species-level differences, aiming to identify potential predictive modulators of severity in the context of primary dengue infection.

## Identification of core and Severity-Specific unique microbial species during dengue

To assess whether the microbiome dynamics contributed to the dengue disease severity and if there are microbial species that are differentially abundant or unique across the mild, moderate, and severe groups, we conducted further analysis of blood microbial species using the

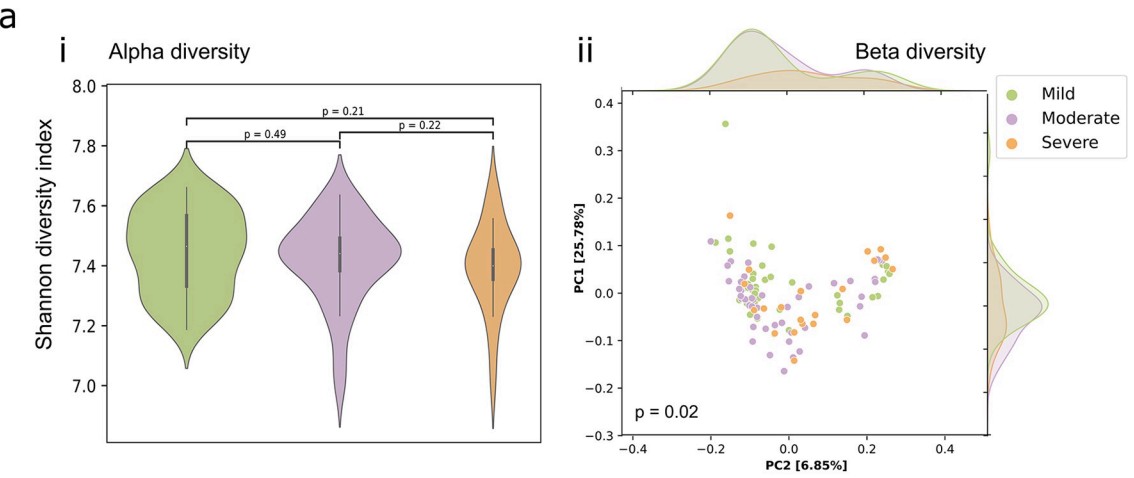

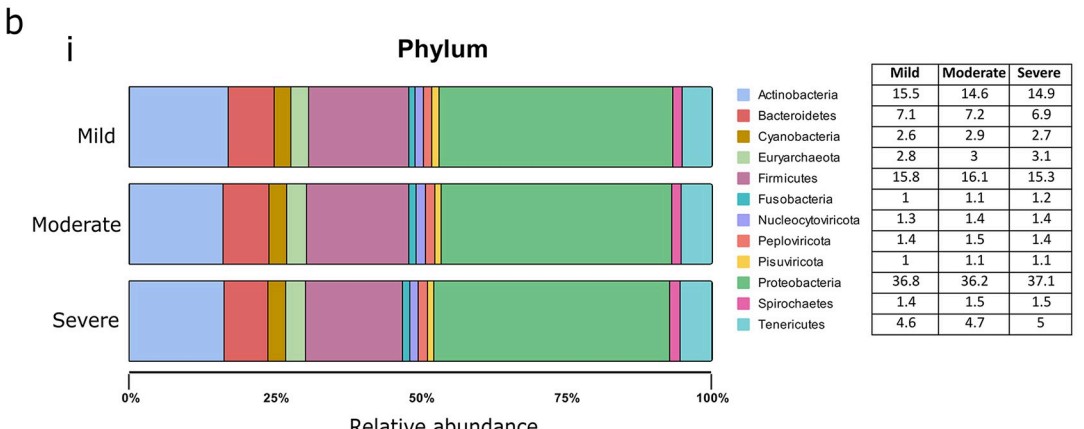

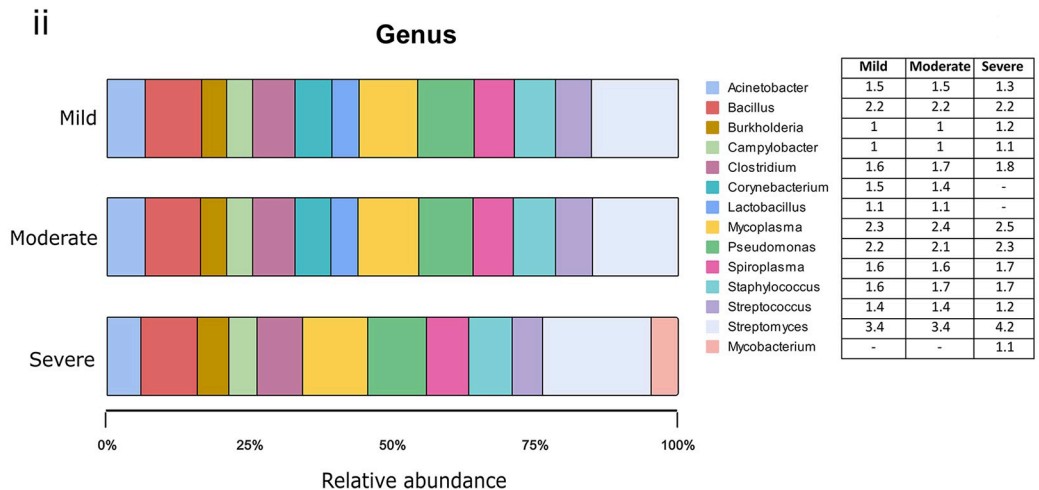

**Fig 3. Microbial Abundance and Diversity in Dengue Patients. (a)** Overview of microbial diversity measures: **i)** Shannon alpha diversity, indicating within-sample diversity, **ii)** Beta diversity, representing between-sample diversity across mild, moderate, and severe Dengue patient groups. **(b)** Distribution of microbial relative abundance (>1%): **i)** Proportions at the phylum level, and **ii)** Proportions at the genus level.

### a    Total TAMs

### b    Significant Core Species

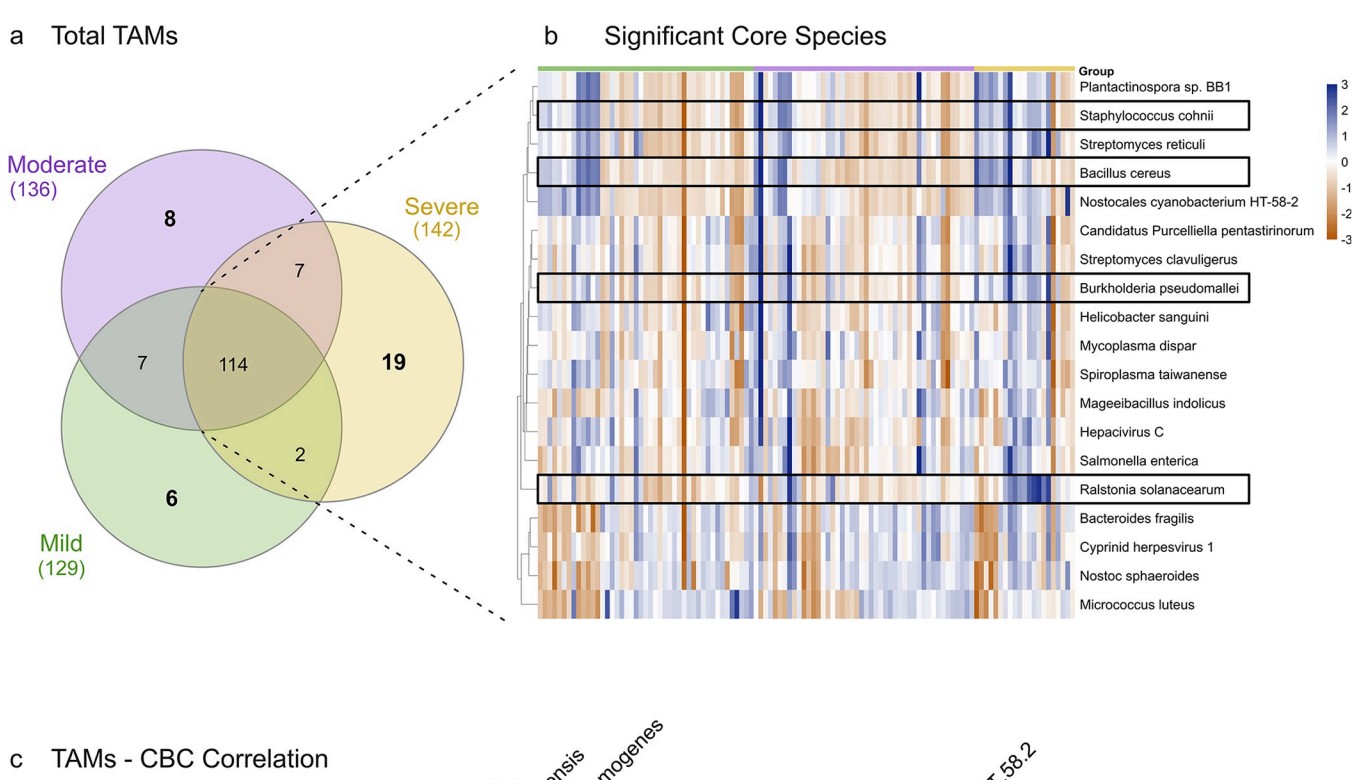

### c    TAMs - CBC Correlation

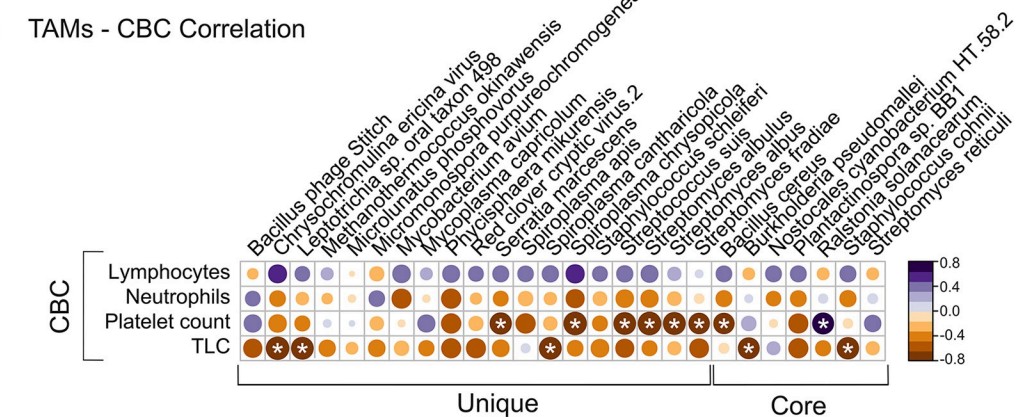

**Fig 4. Microbial species profile across mild, moderate, and severe dengue patients. (a)** Spectrum of microbial species observed across the mild, moderate, and severe Dengue patients. **(b)** The 19 significant core TAMs that differentiate across the severity groups are shown as a heatmap. Box highlights the opportunistic TAMs, significantly abundant in the severe patients. **(c)** Correlation matrix between common and unique TAMs and significant CBC parameters. Significant correlations are highlighted by white stars.

RNA-seq datasets. A total of 129, 136, and 142 microbial species in the mild, moderate, and severe groups, respectively, was captured during analysis with the prevalence in at least 50% of the samples within each severity group and a relative cumulative abundance of 0.1%. From the entirety of species identified, a noteworthy observation was the consistent presence of 114 species across all severity classes, indicating their persistent association with Dengue infection and thus designating them as core microbes. Subsequently, the mild, moderate, and severe groups were observed to harbor 6, 8, and 19 unique species, respectively (**Fig 4A**).

Examining the significant differential abundance of the core TAMs, we utilized STAMP analysis, identifying 19 bacterial species across the groups (**Fig 4B**). It highlights the inter-individual variability in species abundance among dengue patients across mild, moderate, and severe groups. Importantly, upon examining the 19 microbial species identified, we noted that

some species were significantly more abundant in the severe patients compared to the mild and moderate and were reported as opportunistic pathogens. These species include *Burkholderia pseudomallei*, *Staphylococcus cohnii*, *Bacillus cereus* and a harmful plant pathogen, *Ralstonia solanacearum*. On the other hand, the mild group was significantly enriched with both beneficial and opportunistic species, including *Streptomyces clavuligerus*, *Spiroplasma taiwanense*, *Salmonella enterica*, and *Micrococcus luteus*, along with two viruses, *Hepacivirus C* and *Cyprinid herpesvirus 1*. In contrast, *S. enterica* is a major cause of bloodstream infections, particularly in children in developing countries [26], and *M. luteus*, while generally a non-pathogenic skin commensal bacterium, can act as an opportunistic pathogen and cause serious infections.

We observed distinct species exclusively present in each severity group of dengue patients that might be an indicative of dengue disease severity. Specifically, 6 unique species were identified in the mild, 8 in the moderate, and 19 in the severe, indicating an increased microbial diversity associated with greater disease severity. Furthermore, analysis of the mild patients revealed the presence of both commensal and opportunistic species. Of note, *Kocuria* species (*K. palustris* and *K. flava*), typically considered normal skin flora and non-pathogenic to humans, were present. However, these species have recently been associated with opportunistic infections in the immuno-compromised patients [27], suggesting their potential role in mild dengue cases. Additionally, other gram-negative species such as *Neisseria meningitidis*, *Pseudomonas fluorescens*, and *Brucella melitensis* were identified. *N. meningitidis*, although a common nasopharyngeal commensal, is a known pathogen that can cause severe infections under certain conditions [28]. *P. fluorescens*, a less virulent, environmental bacterium, has been implicated in respiratory infections [29]. *B. melitensis*, primarily an animal pathogen, can cause brucellosis in humans, characterized by fever and malaise [30]. Similarly, the moderate patients exhibited the presence of beneficial gram-positive bacteria, such as *Streptococcus thermophilus*, known to promote the growth of healthy intestinal flora [31]. Additionally, gram-negative opportunistic species like *Acinetobacter lwoffii* and *Mannheimia haemolytica* were also identified, potentially reflecting moderate disease severity. Severe patients demonstrated a high diversity of bacterial species, predominantly opportunistic pathogens. These included *Staphylococcus schleiferi*, *Serratia marcescens*, *Streptococcus suis*, *Mycoplasma capricolum*, and *Mycobacterium avium*. Furthermore, several viruses such as *Bacillus Phage Stitch*, *Chrysochromulina ericina*, *Red clover cryptic virus 2*, and an archaeon, *Methanothermococcus okinawensis*, were also found in the severe patients.

The notable alteration in the microbiome, indicative of increased vulnerability among dengue patients, and the diversity of opportunistic pathogens observed in severe cases underscore the potential role of specific microbial signatures in modulating disease severity. For better generalization of severe-group bacterial features (unique and common), we studied whether the significant indicators of clinical blood and liver parameters were related to the microbial community structure in these patients. From clinical indices of CBC and LFT, we found that the TLC (2.7), platelet counts (125), neutrophils (62.6) and total protein (6.7) were significantly decreased in the severe group, while lymphocytes (26), bilirubin (total and direct) (0.8 and 0.2) and SGOT (97.5) were significantly higher (**Fig 2**). Pearson correlation analysis was used to verify the relationship between the bacterial normalized counts and individual specific clinical factors. We utilized a correlation matrix to show the association between 26 microbial species of the severe group (19 unique and 7 common) and the 8 selected clinical factors (**S2 File**). The opportunistic species, *B. cereus*, *B. pseudomallei*, *S. cohni* from the common batch and *S. marcescens*, *S. suis*, along with the species from Spiroplasma and Streptomyces genera were significantly and negatively related to TLC and platelet counts. Notably, none of the species showed significant correlation with the neutrophils and lymphocytes (**Fig 4C**). It is worth

noting that, although altered, neutrophils, lymphocytes, total protein and bilirubin fell in the normal clinical range. Thus, the abnormalities in blood and liver function in severe patients were characterized by TLC and platelet counts and SGOT (AST) respectively. It should be mentioned that among the severe group, only 11 patients had LFT information, which is rate-limiting for robust statistical analysis. Therefore, the observed correlations between liver dysfunction and bacterial presence, such as *S. apis* significantly positively relate to bilirubin and the associations of *S. cohni* and *S. chrysopicola* with SGOT, cannot be stated with confidence (**S3 Fig**). These findings suggest that these species may serve as microbial markers for evaluating disease severity in dengue patients. These insights into host-microbiome interactions highlight the influence of specific microbial profiles on the progression and severity of dengue, offering new perspectives on clinical outcomes.

## Transcriptomic Repertoire of blood microbiome in severe Dengue patients enriched for essential cellular functions and metabolic pathways

**Identification of microbe-specific genes.** Using our dual RNA-seq data, we investigated gene enrichment in the unique TAMs associated with varying dengue severity levels (mild, moderate, and severe). BLASTX analysis against the Uniprot database, coupled with stringent filtering criteria (e-values < 1e-50, no gap openings, minimum percent identity of 50%, and eliminating duplicate genes), enabled the identification of unique genes in microbial species across the severity groups. This comprehensive approach highlighted bacterial adaptive mechanisms in diverse host micro-environments.

Subsequently, we retrieved specific genes corresponding to unique species of mild, moderate and severe patients as presented in **Table 3**. Briefly, in the mild dengue patients, we identified five unique species with notable gene counts: *B. melitensis* (15 genes), *K. flava* and *K. palustris* (7 genes each), *N. meningitidis* (15 genes), and *P. fluorescens* (13 genes). The moderate patients revealed eight unique species, with *M. haemolytica* (24 genes) and *A. lwoffii* (21 genes) being the most prominent. The severe patients included *M. avium* and *S. marcescens* (24 genes each), and *S. suis* (19 genes), highlighting a higher gene count and diversity. Other species such as *M. okinawensis* and *S. albulus* (7 genes each) also contributed to the severe patient's gene profile.

**Functional metabolic pathways enrichment.** We continued our analysis by conducting pathway functional enrichment analysis using ShinyGO (version 0.80) to explore functional significance of genes from unique species across mild, moderate, and severe dengue patients (**S3 File**). A two-step filtering process was employed: enrichment threshold with a false discovery rate (FDR) of ≤ 0.05, and ensuring at least 10% of pathway-associated genes were present in our dataset. The analysis identified distinct numbers of genes linked to metabolic pathways enriched in each severity group (**Fig 5A and Table 3**).

At the functional level, we identified various metabolic pathways with comparable levels in mild and moderate patients, but increased abundance in the severe patients. A Sankey plot (**Fig 5B**) illustrates the genes related to the specific pathways and their severity subgroup presence. Overall, most pathways, including ATP formation, nucleotide biosynthesis, ion transport, cell shape and integrity, protein degradation and stress response were common to all the groups but involved different genes (**Fig 5B**). These pathways highlight cellular energy production, maintenance of cellular integrity, and bacterial adaptability to diverse ecological niches.

In the mild patients, we identified 16 unique genes across all species associated with 11 significant pathways. Species-level associations revealed diverse enrichments: *B. melitensis* showed enrichment in protein quality control (*LON, RPLC, DNAK*), amino acid biosynthesis (*ARGG*), ATP formation (*ATPA, ATPD*), and ion transport (*ATPA, ATPD*). *K. flava* and *K.*

**Table 3. Unique TAMs and associated genes involved in pathways across dengue severity groups.**

| Severity Group | Unique Species | No. of Genes identified by BLAST | Genes involved in Pathways | Enriched Metabolic Pathways |
|---|---|---|---|---|
| Mild | *Neisseria meningitidis* | 15 | 11 | Antioxidant defense, Stress response, NADP binding |
| Mild | *Brucella melitensis* | 15 | 7 | Protein quality control, Amino acid biosynthesis, ATP formation, Ion transport, Stress response |
| Mild | *Pseudomonas fluorescens* | 13 | 8 | Protein synthesis under stress, ATP formation, Carbon metabolism |
| Mild | *Kocuria palustris* | 7 | 5 | ATP formation, Carbohydrate metabolism, Nucleotide biosynthesis, Protein degradation, Stress response |
| Mild | *Kocuria flava* | 7 | 7 | ATP formation, Carbohydrate metabolism, Nucleotide biosynthesis, Protein degradation, Stress response |
| Mild | *Ornithinimicrobium sp AMA3305* | - | - | |
| Moderate | *Mannheimia haemolytica* | 24 | 8 | Protein quality control, Amino acid biosynthesis, Carbohydrate metabolism, NADPH production, Nucleotide biosynthesis, Stress response |
| Moderate | *Vibrio sp. 2521–89* | 22 | - | |
| Moderate | *Acinetobacter lwoffii* | 21 | 12 | Protein quality control, Amino acid biosynthesis, ATP formation, Ion transport, Lipopolysaccharide biosynthesis, NADPH production, Nucleotide biosynthesis, Protein synthesis, Stress response |
| Moderate | *Streptomyces sp. RTd22* | 15 | - | |
| Moderate | *Mycolicibacterium chubuense* | 10 | - | |
| Moderate | *Anabaena sp. 90* | 8 | 8 | ATP formation, Carbohydrate metabolism, Cell shape, Ion transport, NADPH production, Nucleotide biosynthesis, Protein quality control, Stress response |
| Moderate | *Streptococcus thermophilus* | 5 | 5 | Protein quality control, ATP formation, Defence against oxidative stress, Ion transport, Nucleotide biosynthesis, Protein synthesis, Stress response |
| Moderate | *Human endogenous retrovirus K* | 2 | - | |
| Severe | *Serratia marcescens* | 24 | - | |
| Severe | *Mycobacterium avium* | 24 | 3 | ATP synthesis, Ion transport, Nucleotide synthesis |
| Severe | *Streptococcus suis* | 19 | 2 | ATP synthesis, Ion transport, Nucleotide synthesis |
| Severe | *Streptomyces albulus* | 7 | 3 | ATP synthesis, Ion transport, Nucleotide synthesis, Stress response |
| Severe | *Methanothermococcus okinawensis* | 7 | 2 | ATP synthesis, Ion transport, Nucleotide synthesis, Protein degradation |
| Severe | *Streptomyces fradiae* | 6 | 2 | ATP synthesis, Cell shape and integrity, Ion transport, Nucleotide synthesis, Stress response |
| Severe | *Leptotrichia sp. oral taxon 498* | 5 | - | |
| Severe | *Streptomyces albus* | 5 | 3 | ATP synthesis, Ion transport, Nucleotide synthesis, Stress response |
| Severe | *Staphylococcus schleiferi* | 5 | 2 | ATP synthesis, Ion transport, Nucleotide synthesis |
| Severe | *Phycisphaera mikurensis* | 4 | - | |
| Severe | *Spiroplasma apis* | 3 | - | |
| Severe | *Spiroplasma cantharicola* | 3 | | |
| Severe | *Spiroplasma chrysopicola* | 3 | - | |
| Severe | *Mycoplasma capricolum* | 3 | | |
| Severe | *Micromonospora purpureochromogenes* | 2 | - | |
| Severe | *Microlunatus phosphovorus* | 2 | - | |
| Severe | *Chrysochromulina ericina virus* | 1 | - | |
| Severe | *Bacillus phage Stitch* | - | - | |
| Severe | *Red clover cryptic virus 2* | - | - | |

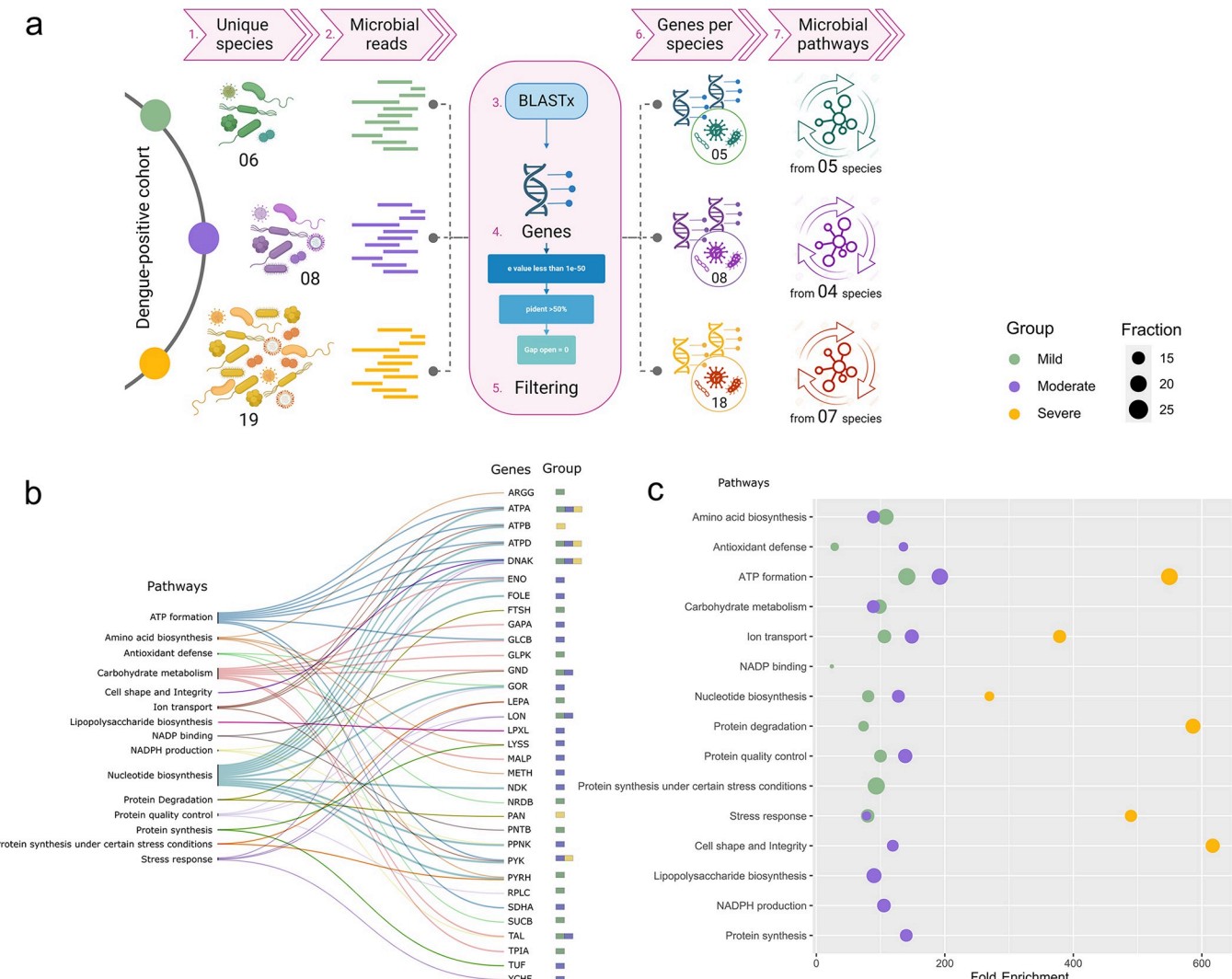

**Fig 5. Microbial genes and pathways.** (a) Schematic diagram illustrating the workflow for identifying unique microbial species genes in mild, moderate, and severe dengue patients using BLASTX and pathways using Shiny GO. Figure created with Biorender.com. (b) Alluvial plot depicting pathways associated with microbial genes, where their presence is visualized with distinct colors. (c) Bubble plot displaying enriched pathways across mild, moderate, and severe groups of dengue patients, highlighting their significance and distribution.

*palustris* exhibited enrichment in ATP formation, carbohydrate metabolism (*GLPK*), nucleotide biosynthesis (*PRS*), and stress response (*DNAK*). Additionally, the opportunistic microbes, *N. meningitidis* and *P. fluorescens* demonstrated pathways related to antioxidant defence (*SUCB*, *NRDB*), stress response (*LON*, *DNAK*), and NADP binding (*GND*, *PNTB*). A total of 21 genes were involved in 12 major pathways in the moderate patients, wherein *A. lwoffii*, *S. thermophilus*, *M. haemolytica*, and *A. sp. 90* displayed involvement in a wide range of pathways including protein quality control, amino acid biosynthesis, and stress response, indicating versatile metabolic adaptations. Key genes involved in ATP formation (*ATPA*, *ATPD*), carbohydrate metabolism (*TAL*, *GLCB*), and nucleotide biosynthesis (*PPNK*) were identified. Of the 19 species in the severe patients, 7 TAMs displayed distinctive genes enriched within significant pathways. However, across all species in the severe group, there were merely 6 unique genes associated with 6 significant pathways, specifically stress response, protein degradation, nucleotide biosynthesis, ion transport, cell shape and integrity, and ATP formation

(**Fig 5B**). Upon examining each species, it was observed that *S. albus* and *S. albulus* share pathways such as ATP synthesis (*ATPA*, *ATPD*), ion transport (*ATPA*, *ATPD*), and nucleotide synthesis (*ATPA*, *ATPD*), indicating a common metabolic profile. *S. schleiferi* and *S. suis* also showed similar pathways, with the presence of multiple ATP synthesis genes, suggesting a high energy demand in these species.

It's worth noting that certain pathways, including protein quality control, antioxidant defence, carbohydrate metabolism and amino acid biosynthesis, were exclusively found in mild and moderate cases, but not in severe patients (**Fig 5C**). In bacteria, protein quality control mechanisms are vital for maintaining proteostasis, ensuring proper protein folding, and preventing the accumulation of damaged proteins. Proteases like Lon, ClpXP, and FtsH play crucial roles in this process. Antioxidant defence pathways, prominent in mild cases, help counteract oxidative stress by scavenging reactive oxygen species (ROS) and maintaining cellular homeostasis. These pathways collectively signify the adaptability of bacteria in milder conditions, emphasizing the importance of protein quality control and antioxidant mechanisms in maintaining bacterial health and functionality. The primary aim of these pathways is to prevent the accumulation of ROS, which can lead to multiple health issues, and damaged proteins notably observed in the mild group [32]. Moreover, the presence of genes ArgG and MetH in bacterial amino acid biosynthesis pathways underscores their essential role in bacterial physiology, including protein synthesis, growth, stress responses, and interactions with hosts and other microbes, especially in the mild and moderate patients. However, the enrichment scores of all functional pathways were higher in the severe patients compared to mild and moderate (**Fig 5C**). This suggests that bacteria under stress conditions often require increased ATP formation, amino acid biosynthesis, and nucleotide biosynthesis to support cellular responses and adaptation. Stress response pathways also indicate the ability of bacteria to thrive in diverse ecological niches.

## Discussion

Our interest in conducting this study stemmed from the existing literature indicating alteration in the microbial composition of the *Aedes aegypti* and *Aedes albopictus* mosquitoes, key vectors for dengue transmission. The endosymbiont bacteria Wolbachia has been shown to inhibit flaviviruses in mosquitoes, including dengue virus (DENV) and Zika virus (ZIKV) [33]. Recently, a study discovered that a naturally isolated symbiotic bacterium, *Rosenbergiella_YN46*, resides in the gut of Aedes mosquitoes and protects them from flavivirus infection by acidifying the insect gut lumen [34]. An effort to understand the functional human microbiome could offer new insights into the prevalence of such microbial species, which may either protect against DENV infection or increase susceptibility to the virus. This can offer an effective and safe approach to DENV control, especially when there is no preferred current treatment or vaccine available against the virus. Besides the known role of microbiota in human health and infectious diseases, including the COVID-19 pandemic [35–37], no study on human microbiota in dengue disease, especially different severities has been conducted. We therefore aimed to identify, for the first time, the early blood microbial signatures following dengue virus infection in patients with varied dengue disease severity using a transcriptomics-based approach.

The fact that NS1 Ag of DENV circulates to high levels in the bloodstream of patients prompted us to explore microbial dynamics in blood as a potential modulator of dengue's clinical diversity [38]. While the notion of blood microbiome remains controversial, researchers have evidenced microbial signatures in blood that are altered in various disease states such as cardiovascular diseases, chronic inflammatory diseases, and cancer, possibly reflecting

translocation from other body sites (gut or oral cavity) or altered immune interactions [39,40]. Upon analyzing the taxonomic composition at the phylum and genus levels, all three severity groups exhibit a dominance of Proteobacteria (pathogenic bacteria of blood microbiome), followed by Firmicutes and Actinobacteria, consistent with our previous research on the dengue serum microbiome [8]. Additionally, a study on gut translocation biomarkers also revealed the enhanced presence of Proteobacteria in the critical dengue cases [41]. Proteobacteria, especially genera such as Burkholderia, Campylobacter, and Pseudomonas, were consistently present, along with Firmicutes like Bacillus, Clostridium, Staphylococcus, and Streptococcus, and Actinobacteria such as Corynebacterium and Streptomyces.

Markedly, the identification of 19 statistically significant core TAMs within the cohort reveals the prevalence of specific pathobiont species such as *B. pseudomallei*, a gram negative bacterium known to cause acute fatal sepsis in tropical and subtropical regions, which has been previously detected in dengue patients of North-eastern Brazil, leading to severe co-infections. Other opportunistic pathogens, including *B. cereus*, *S. cohnii*, *and R. solanacearum*, *were also identified* in severe cases [42]. The pathogenic *B. cereus* causes a number of systemic and local infections in both immunologically compromised and immunocompetent individuals [43]. Similarly, *S. cohnii*, though infrequent, poses significant risks due to its implification in urinary tract infections, bacteremia, and sepsis, particularly in immunocompromised and hospitalized individuals [44]. Our identification of the unique species captures the coexistence of diverse opportunistic microbes, providing insights into increased severity observed in dengue patients. Chief among our findings is the presence of *S. marcescens*, an opportunistic gram-negative bacterium commonly associated with nosocomial infections, including pneumonia and bloodstream infections, with high mortality rates, especially in immunocompromised patients [45]. Interestingly, Wu et. al. identified *S. marcescens* as a mosquito gut commensal that enhances the susceptibility of mosquitoes to dengue virus via a secreted protein [46]. The corroborating presence of this bacterium in our cohort of severe dengue patients might suggest its potential role in promoting DENV infection. Additionally, the severe group showed an increased abundance of *S. schleiferi*, *S. suis*, *M. capricolum*, and *M. avium*. *M. avium* is known for causing severe lung infections, particularly in immunocompromised individuals, and it is a common pathogen in patients with HIV/AIDS [47]. It can also cause disseminated infections, including bloodstream infections, leading to sepsis. Although rare in human infections, *M. capricolum*, has been associated with respiratory infections, often in individuals with underlying health conditions and has been implicated in bloodstream infections, particularly in immunocompromised patients [48]. *S. suis* causes severe systemic infections like meningitis and septicemia, often transmitted through contact with infected pigs or pork products [49]. Given the associations of these opportunistic species with bloodstream infection and septicemia, our findings of their presence in the blood microbiome are interesting, underscoring their potential contribution to increased disease severity in dengue patients. *S. schleiferi*, although uncommon in humans, has been linked to various conditions, including surgical site infections and pediatric meningitis, producing virulence factors (such as beta-hemolysin, lipase, and esterase) that contribute to its pathogenic potential in causing skin and soft tissue infections in humans and animals [50]. The Streptomyces species identified in the severe cases (*S. abulus*, *S. albus*, and *S. fradie*) are known for their antibacterial activity, achieved through the synthesis of specific bioactive molecules [51,52]. These species may potentially aid patients in combating the infection, thereby preventing further progression to more severe conditions such as liver damage or bleeding in dengue-positive patients. In contrast, mild dengue patients exhibited a reduced presence of opportunistic species. The notable increase in bacterial diversity, based on transcriptional activity, observed in the severe patients compared to mild and moderate disease, suggests a potential role of dysbiotic microbiomes in disease severity.

It is well known that dengue is clinically heterogeneous, presenting with symptoms ranging from asymptomatic to life-threatening cases, depending on infecting serotype, age, immunity, and whether it is a primary or secondary infection. Encouragingly, we identified negative correlations of platelet and total leukocyte counts with the severe-cases species. Specifically, *B. pseudomallei* and *S. cohnii* were negatively correlated with TLC, while *B. cereus*, *S. marcescens*, *S. abulus*, *S. albus*, and *S. fradie* were negatively correlated with platelet counts. The enzyme bacethrombase, purified from *B. cereus*, has demonstrated antiplatelet activity, suggesting its potential impact on dysregulated platelet counts in severe dengue cases [53]. *B. pseudomallei*'s high rate of latent infection is linked to its ability to survive and multiply within human leukocytes, thereby evading the host immune response [54]. The negative correlation between this pathogen and TLC warrants further exploration. Given the association of potential pathogens like *B. cereus*, *S. cohnii*, and *B. pseudomallei* with key indicators of dengue severity, these species may serve as microbial markers for evaluating the severity of dengue fever. Detailed characterization and identification of microbial features linked to these infections could facilitate early detection, enabling timely and targeted treatment. Targeting these specific microbes presents a promising approach for early diagnosis, offering a non-invasive and potentially more accurate method for identifying health conditions at their earliest stages. As ours' is the first study elucidating blood microbiome of Dengue patients with different disease severities, future metatranscriptomic studies in this direction from different cohorts within India as well as globally would be great value addition. Finally, we provide functional evidence of dengue-associated TAMs by utilizing microbial sequencing reads to identify the genes and pathways associated with unique bacteria identified within each severity group. Pathway analysis helped us to identify gene expression patterns and pinpoint which bacteria are actively replicating and potentially transcribing their genes. It is important to note that microbial transcriptional responses can quickly adjust to changes in environmental conditions, such as fluctuations in inflammation and oxygen levels [55]. Interestingly, we identified that the severe cases exhibited enrichment in pathways crucial for cellular energy production and cellular integrity maintenance, such as nucleotide biosynthesis, ion transport, and ATP formation, with only 5 associated genes (*ATPA*, *ATPD*, *ATPB*, *PYK*, *DNAK*). This highlights the bacteria's adaptability to diverse ecological niches and their ability to thrive, contributing to their ecological success and resilience. These enrichment analyses were performed in an exploratory manner; we acknowledge the need for a more systematic approach to validate these findings.

Since these studies are rare, the range of microbial dynamics in dengue infection, more so with different disease severity, is just beginning to be explored. One limitation of our study is that it remains unclear whether these microbes are present at the onset of dengue fever. Further research is needed to determine the timing of their appearance/significant differential abundance in relation to the disease's progression, which would be essential for evaluating their potential as early diagnostic markers. Insights would be strengthened with longitudinal studies of the Dengue patients to understand the microbial dynamics. In total, our blood microbiome study represents a preliminary exploration of the early opportunistic TAMs modulating clinical parameters and their enriched metabolic pathways associated with the genes that govern the establishment of severe infection (**Fig 6**). With our initial set of findings revealing an increased diversity of TAMs in the severe disease patients, it will be interesting to see further insights into the human microbiome that could be translated to therapeutic strategies in DF.

## Conclusions

The role of microbiota has been extensively studied in infectious diseases, but little is known about its impact on dengue patients. By employing dual RNA-seq data, we provide new

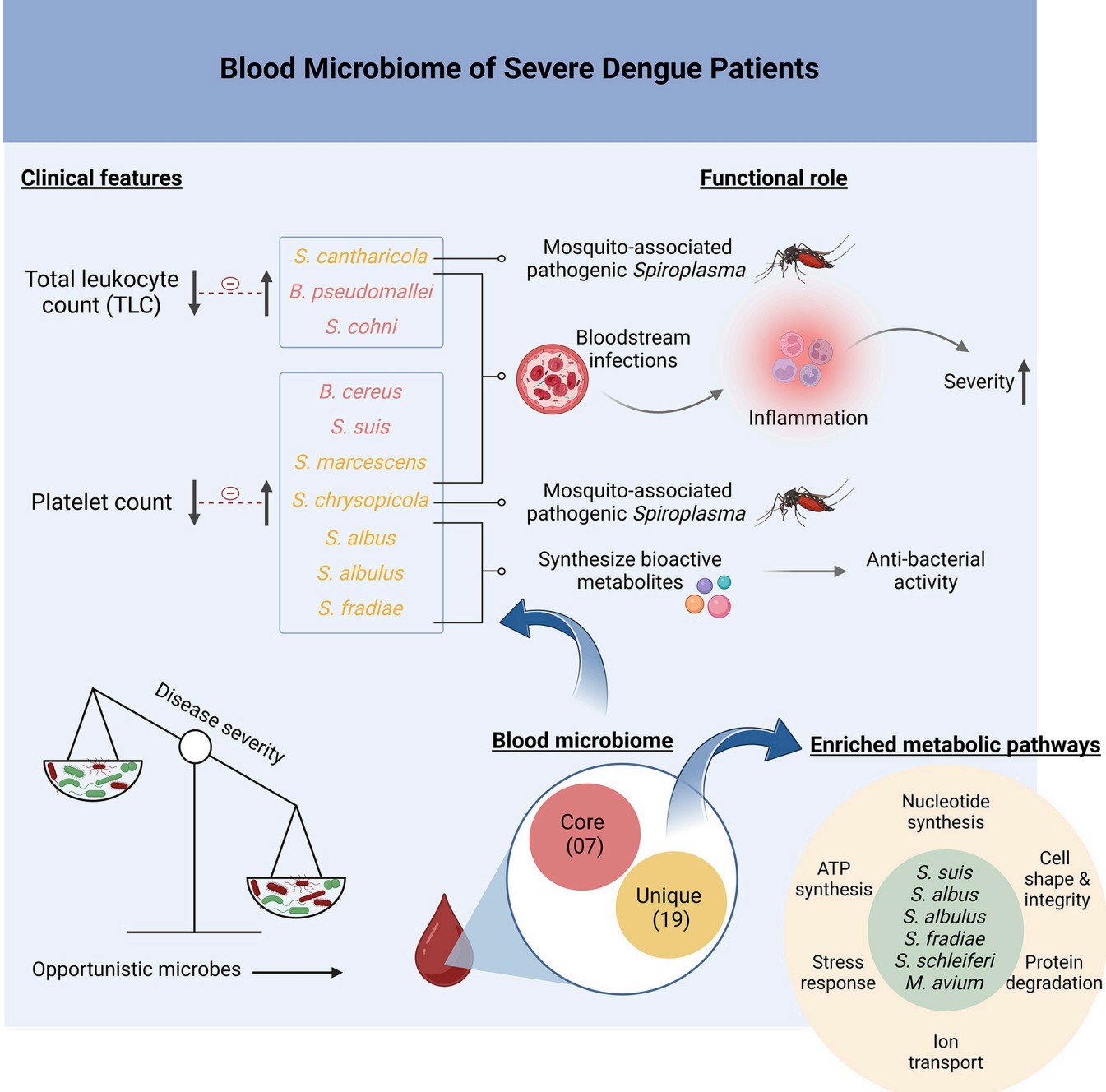

**Fig 6. Functional Role of Specific Opportunistic Blood Transcriptionally Active Bacterial Species in Dengue Disease.** This Figure highlights the correlation between the increased presence of specific transcriptionally active microbial species in the bloodstream and the possible exacerbation of inflammatory responses and enriched metabolic pathways, which are associated with enhanced disease severity in the dengue patients. Figure created with Biorender.com.

insights into the blood microbial profile of the dengue-positive patients. To our knowledge, this is the first study generating the human blood microbiome dataset for dengue infection, revealing the unexplored landscape of TAMs in the severe patients. Our findings highlight an increased abundance of opportunistic TAMs among the severe dengue patients, with few of

them showing significant correlations with platelet counts and TLC. The enriched metabolic pathways observed in the severe patients evidenced by our study suggest heightened metabolic activity of these opportunistic species. Hence, investigating the diseases-associated microbial signatures at species-level will contribute to enhanced understanding of the microorganisms involved in dengue infection. Additional research into whether manipulation of microbial populations can ameliorate disease holds the potential for advancing clinical care. Therapeutic manipulation of the microbiota could become a valuable strategy for the prevention and treatment of DF.

## Supporting information

**S1 File. Phylum and genus level taxonomic communities across dengue severity groups.**
(XLSX)

**S2 File. Pearson correlation coefficients (r) and p values obtained between TAMs of severe group and significant clinical parameters (CBC and LFT).**
(XLSX)

**S3 File. Functional pathways along with associated genes across mild, moderate and severe groups of Dengue patients via ShinyGO.**
(XLSX)

**S1 Fig. Venn diagram illustrating the overlapping and unique symptoms across the severity groups.**
(PNG)

**S2 Fig. Simpson and Chao1 alpha diversity indices.**
(TIF)

**S3 Fig. Pearson correlation analysis between TAMs and LFT parameters.**
(TIF)

## Acknowledgments

Authors would like to acknowledge all the dengue patients who participated in the study. Authors duly acknowledge the help and support from Dr. Aradhita Baral as a research manager and Dr. Bharti Kumari as lab manager, towards facilitation and coordination with the funders. Authors further recognise P.L. Bhatt for his contribution to providing samples. Authors acknowledge the support of Anil Kumar and Nisha Rawat for their assistance with sample transport and management.

## Author Contributions

**Conceptualization:** Rajesh Pandey.

**Data curation:** Pallawi Kumari.

**Formal analysis:** Aanchal Yadav, Priti Devi, Pallawi Kumari.

**Funding acquisition:** Rajesh Pandey.

**Investigation:** Aanchal Yadav, Priti Devi, Pallawi Kumari, Uzma Shamim, Rajesh Pandey.

**Methodology:** Aanchal Yadav, Priti Devi, Uzma Shamim.

**Project administration:** Rajesh Pandey.

**Resources:** Bansidhar Tarai, Sandeep Budhiraja, Rajesh Pandey.

**Supervision:** Rajesh Pandey.

**Visualization:** Aanchal Yadav, Priti Devi, Pallawi Kumari.

**Writing – original draft:** Aanchal Yadav, Priti Devi, Pallawi Kumari, Uzma Shamim.

**Writing – review & editing:** Rajesh Pandey.

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
