## [Decision Letter · Decision Letter 0]

26 Aug 2024

Dear Dr. Pandey,

Thank you very much for submitting your manuscript "Metagenomic insights into the dengue patient blood microbiome: Enhanced microbial diversity and metabolic activity in severe patients" for consideration at PLOS Neglected Tropical Diseases. As with all papers reviewed by the journal, your manuscript was reviewed by members of the editorial board and by several independent reviewers. The reviewers appreciated the attention to an important topic. Based on the reviews, we are likely to accept this manuscript for publication, providing that you modify the manuscript according to the review recommendations.

Sincerely,

Antonio Mas

Academic Editor

Andrea Marzi

Section Editor

Dear Dr. Rajesh Kumar Pandey,

We sent your manuscript to two reviewers and both agree in their assessments, minor revisions. The comments are below this email. Please read these comments carefully and respond to them.

Best regards,

Antonio Mas

Reviewer's Responses to Questions

**Key Review Criteria Required for Acceptance?**

**Methods**

-Are the objectives of the study clearly articulated with a clear testable hypothesis stated?

-Is the study design appropriate to address the stated objectives?

-Is the population clearly described and appropriate for the hypothesis being tested?

-Is the sample size sufficient to ensure adequate power to address the hypothesis being tested?

-Were correct statistical analysis used to support conclusions?

-Are there concerns about ethical or regulatory requirements being met?

Reviewer #1: The study design is robust, and the statistical analyses used are appropriate to support the conclusions. There are no concerns regarding ethical requirements. Although the sample size is limited, it is sufficient to substantiate the conclusions.

Reviewer #2: Yes

**Results**

-Does the analysis presented match the analysis plan?

-Are the results clearly and completely presented?

-Are the figures (Tables, Images) of sufficient quality for clarity?

Reviewer #1: The analysis presented is well-suited to address the research question, and the results are clearly articulated. The figures are of high quality.

Minor Comments:

• Table 1: Include dispersion values in the data.

• Table 2: Clarify that these are metabolic pathways.

• Page 25: Italicize "S. marcescens."

• Page 26: Remove italics in "although uncommon in humans."

Reviewer #2: Yes

**Conclusions**

-Are the conclusions supported by the data presented?

-Are the limitations of analysis clearly described?

-Do the authors discuss how these data can be helpful to advance our understanding of the topic under study?

-Is public health relevance addressed?

Reviewer #1: The conclusions are well-supported by the data, and the study's limitations are adequately described. The implications for public health are addressed, and the relevance of the work is clearly articulated.

Minor Comments:

• Page 26: The statement "these species may serve as microbial markers for evaluating the severity of dengue fever" is significant. I suggest elaborating on this idea, emphasizing the potential for early diagnosis. Additionally, consider mentioning as a limitation that it is unclear whether these bacteria are present at the onset of the disease.

• Figure Legend 6: The phrase "active microbial species in the bloodstream exacerbates inflammatory responses and enriched metabolic pathways" implies causation, while the study only shows correlation. The main text does not exhibit this overinterpretation.

Reviewer #2: Yes

**Editorial and Data Presentation Modifications?**

Reviewer #1: (No Response)

Reviewer #2: (No Response)

**Summary and General Comments**

Reviewer #1: Summary and General Comments

This study is highly compelling, linking the blood microbiome of dengue patients with biometric data on the inflammatory response. The use of RNA-seq to confirm that the bacteria are metabolically active is commendable, and the correlations found between certain species and metabolic pathways with disease severity are significant. The study is well-executed, employing appropriate methodologies and statistical analyses.

I have only a few minor comments, as noted in the Results and Conclusions sections. It was a pleasure to review this manuscript.

Reviewer #2: Yadav et al. report the microbial diversity in dengue patients. Dengue is an emerging problem that is associated with climate change and is becoming an increasing problem in India. The study is straightforward and uses a rigorous data analysis pipeline to report on the microbial diversity in clinically relevant specimens. The study is informative, and I only have minor comments that are appended below.

1. Figure 2A – how many patients had overlapping symptoms? It will be useful to represent the data as overlapping Venn Diagrams to illustrate the overlapping and unique symptoms in the patient population. 

2. How does the data compare to healthy individuals? While the comparison of mild, moderate, and severe cases is very relevant and important, including healthy controls will make the data more meaningful and potentially enable the discovery of Dengue-related biomarkers and microbiome disruption, if any.

3. Please submit all raw data, analysis codes and scripts in a public repository, and include a freely accessible link to the code and data in the manuscript. Eg., GitHub, NCBI.

PLOS authors have the option to publish the peer review history of their article (what does this mean?). If published, this will include your full peer review and any attached files.

Reviewer #1: Yes: Mariana Peimbert

Reviewer #2: No

Figure Files:

Data Requirements:

Reproducibility:

References

---

## [Editor Report · Decision Letter 1]

2 Oct 2024

Dear Dr. Pandey,

We are pleased to inform you that your manuscript 'Metatranscriptomic insights into the dengue patient blood microbiome: Enhanced microbial diversity and metabolic activity in severe patients' has been provisionally accepted for publication in PLOS Neglected Tropical Diseases.

Best regards,

Antonio Mas

Academic Editor

Andrea Marzi

Section Editor

<style type="text/css">p.p1 {margin: 0.0px 0.0px 0.0px 0.0px; line-height: 16.0px; font: 14.0px Arial; color: #323333; -webkit-text-stroke: #323333}span.s1 {font-kerning: none

</style>

---

## [Editor Report · Acceptance letter]

11 Oct 2024

Dear Dr. Pandey,

We are delighted to inform you that your manuscript, "Metatranscriptomic insights into the dengue patient blood microbiome: Enhanced microbial diversity and metabolic activity in severe patients," has been formally accepted for publication in PLOS Neglected Tropical Diseases.

Best regards,

Shaden Kamhawi

co-Editor-in-Chief

Paul Brindley

co-Editor-in-Chief
